# Proactive Personality and Construction Worker Safety Behavior: Safety Self-Efficacy and Team Member Exchange as Mediators and Safety-Specific Transformational Leadership as Moderators

**DOI:** 10.3390/bs13040337

**Published:** 2023-04-18

**Authors:** Junwen Mo, Libing Cui, Ruirui Wang, Xuesong Cui

**Affiliations:** Department of Engineering Management, School of Civil Engineering, Lanzhou Jiaotong University, Lanzhou 730070, China11200236@stu.lzjtu.edu.cn (R.W.); 11210235@stu.lzjtu.edu.cn (X.C.)

**Keywords:** proactive personality, safety behavior, safety self-efficacy, team member exchange, safety-specific transformational leadership

## Abstract

Research on the correlation between personality traits and safety behaviors has been thoroughly explored in previous literature. However, most of these studies are based on explaining the relationship between the Big Five personality traits and safety behavior, with few explaining the relationship between proactive personality and safety behavior. This study relies on trait activation theory, social cognitive theory, and social exchange theory to understand the relationship between proactive personality and safety behavior (safety participation and safety compliance) by using safety self-efficacy and team member exchange as mediating variables and safety-specific transformational leadership as moderating variables. Method: Considering the issue of common method bias, a multi-source and multi-stage data collection research design was used to collect 287 valid questionnaires from construction workers in 10 construction projects and apply regression analysis for hypothesis testing. Conclusions: Research results indicated that proactive personality positively and significantly influenced construction workers’ safety behaviors, while safety self-efficacy and team member exchange partially mediated the relationship between proactive personality and safety behaviors. In addition, safety-specific transformational leadership enhanced the positive relationship between proactive personality and safety behavior. These findings enrich the research on the correlation between personality traits and safety behaviors of construction workers in a safety context.

## 1. Introduction

The construction industry is one of the most dangerous industries in the world, and the frequency of accidents, injuries, and fatalities each year takes a huge toll on human life, property, families, and company reputations [1]. According to statistics, in 2021, there were 734 construction safety accidents and 840 fatalities in housing and municipal construction in China, an increase of 42 accidents and 33 fatalities over 2020, up by 6.1% and 4.1%, respectively [2]. Among the many factors that lead to accidents, most involve human factors, with more than 80% of workplace accidents being related to the unsafe behavior of workers [3,4]. Meanwhile, previous studies have shown that differences in individual worker characteristics are an important cause of unsafe behaviors [5]. Among these individual characteristics, personality traits are considered to be an important factor influencing safety behavior [5,6,7].

In recent years, research on the correlation between personality traits and safety behavior has received considerable attention in many fields. However, existing studies have mainly focused on the influence of the Big Five traits (Openness to experience, Extraversion, Agreeableness, Emotional stability, and Conscientiousness) [5,8], but with the in-depth development of personality trait psychology, scholars have noticed that a proactive personality plays a significant role in explaining employee behavior [6,9,10] and has a greater influence than the Big Five personality traits [3,4]. Many studies have investigated the positive effects of proactive personality and related personal outcomes, including attitudes, behaviors, performance, and other career-related outcomes [10,11,12]. However, relatively little research has been done on the correlation between proactive personality and safety behaviors.

Safety behavior is defined by Griffin and Neal [13] as a specific type of job performance that indicates a proactive behavior in which employees take the initiative to follow safety rules and regulations (e.g., safety compliance) and to also help others during job operations (e.g., safety participation). In particular, for construction workers who are direct participants in construction activities, most safety accidents are caused by a relatively small number who violate the rules and regulations of safety behavior and fail to follow them proactively [14]. For example, among 310 workers, Sutherland and Cooper [15] observed that some of them were more prone to accidents than others, indicating that self-management and initiative become very important to improve the level of safety behavior of construction workers. Summarily, this study aims to reveal the mechanism of the effect of proactive personality on safety behavior in the context of safety.

From a relational perspective, proactive personality originates from trait activation theory [16] and is closely related to social cognitive theory [17] and social exchange theory [12]. In this paper, we use these three theories as a general framework to investigate the correlation between proactive personality and safety behavior in the context of construction safety. First, there is self-efficacy. This is a core variable of social cognitive theory. Individuals with high self-efficacy are quite confident in their successful completion of tasks. They have positive attitudes toward challenges in complex environments and are motivated to use effective ways to control behavior [18]. Empirical studies in different fields have shown that self-efficacy has a significant positive effect on behavior and performance [19,20,21], therefore, many scholars have considered self-efficacy to be an intermediate variable between various environmental or individual factors and individual behavior in order to better explain the motivational process of such behavior. [8,22]. For example, the study of Fay and Frese [23] showed that self-efficacy can be used as a mediating variable to influence the relationship between proactive personality and proactive behavior. However, Bulger and Mellor [22] argued that domain-specific, task-specific, and problem-specific self-efficacy are more predictive of behavior or performance than general self-efficacy. Thus, in the context of construction safety, this study proposes that safety self-efficacy may mediate the relationship between proactive personality and safety behavior. Second, there is social exchange theory. This states that the behavior of individuals is influenced by others, such as leaders, and that proactive personality is used to influence employee behavior and performance by facilitating the exchange of leadership members [12]. However, in building construction, workers are usually employed by small subcontractors in teams and so there is more of an exchange relationship between workers. In other words, the degree of influence of workers on other workers’ behavior is greater than the influence of leaders on workers. For example, in team member exchange, Seers [24] indicates “an individual member’s perception of the overall exchange relationship between him and other members of the team”. This relationship should be considered when examining the influence of proactive personality on employees’ behavior [9,25]. Therefore, the present study proposes that team member exchange may mediate the relationship between proactive personality and safety behavior. Third, there is trait activation theory. This states that personality traits require relevant contexts to express trait-related behaviors and research has shown that motivated workers do not always exhibit behaviors that are beneficial to the organization [26]. This brings in safety transformational leadership as a situational factor related to personality traits and responds to recent calls to compare the influence of personal and situational antecedents on safety behaviors and to identify boundary conditions in the relationship between personality and current limited workplace safety [5,14,27]. Therefore, this study suggests that safety-specific transformational leadership may play a moderating role in the influence of proactive personality on safety behaviors.

In summary, this study fills in the gaps in the relationship between proactive personality and safety behavior and poses the following questions: (i) Does proactive personality have an impact on safety behavior? (ii) Do safety self-efficacy and group member exchange mediate the effects of proactive personality on safety behavior? (iii) How does safety-specific transformational leadership mediate the influence of proactive personality on safety behavior? The specific model is shown in Figure 1.

## 2. Theory and Hypotheses

### 2.1. Proactive Personality and Safety Behavior

Safety behavior is defined as all the behaviors of employees during the process of an operation to comply with the code of practice and to be able to protect themselves and use tools and equipment in the event of a safety accident [28]. Research on the component dimensions of safety behavior needs to be traced back to research on the component dimensions of job performance, and as mentioned in the introduction, safety behavior is defined as a specific type of job performance [13]. Lock [29] divided the dimensions of job performance into task performance and relational performance; Neal and Griffin [13] introduced task performance and relational performance into the safety domain-based on studies related to job performance. They divided safety performance into two dimensions at the behavioral level: (i) safety compliance behavior and (ii), safety participation behavior. Safety compliance behavior refers to the core activities that individuals have to perform to maintain workplace safety, including compliance with rules and regulations and operating procedures, which is an in-role behavior. Safety participation behavior refers to individuals’ involvement in activities to improve their safety behavior, such as warning others about violations and helping others to solve safety problems, which is an extra-role behavior. Safety behavior is influenced by a variety of organizational factors (e.g., safety climate [30,31], safety culture [32], and individual factors). For example, these factors could include psychological capital [33] and personality traits [5,34]. In particular, proactive personality traits reflect an individual’s tendency to initiate behavior and are important predictors of safety behavior.

Proactive personality is a unique personality trait that is different from the Big Five traits [35]. It was first introduced by Bateman and Crant [6] and refers to a stable tendency of individuals to be unconstrained by environmental resistance and to influence and change their surroundings. Specifically, workers with high proactive personalities are relatively unconstrained by the forces of their surroundings, and usually have aggressive qualities and higher-value pursuits. They are good at identifying problems and seeking and capturing opportunities. In particular, when suffering from difficulties and facing challenges in building construction jobs, they are often able to actively respond and show a proactive way of working. Not only will they actively comply with the safety regulations, but they will also be willing to participate in developing Safe Work Procedures for construction, thereby fully utilizing their individual characteristics and behavioral tendencies. This helps to improve the level of building construction safety [7,36]. On the contrary, individuals with low proactive personalities face their environment and its changes negatively and may be changed by it, which in turn weakens their motivation to engage in safety compliance and safety participation. Therefore, the following hypotheses are proposed in this study.

**Hypothesis** **1a:**
*Proactive personality is positively associated with safety compliance.*


**Hypothesis** **1b:**
*Proactive personality is positively associated with safety participation.*


### 2.2. The Mediating Role of Safety Self-Efficacy

The self-efficacy concept was introduced by the famous American psychologist Bandura [18]. It means “the belief in one’s ability to organize and execute the plan of action needed to deal with a potential situation”. As a core concept of social cognitive theory, it can be divided into general self-efficacy and specific self-efficacy, with specific self-efficacy having a more significant facilitative role in shaping individuals’ specific behaviors [37]. In the field of safety, safety self-efficacy refers to “an individual’s belief that he or she is capable of performing safety-related tasks”, which is different from safety awareness and is more oriented toward judgment of one’s own abilities [38]. In the workplace, workers with high proactive personalities tend to break the constraints of adverse factors related to work safety, evaluate their environment more positively and optimistically, rate their talents and abilities more highly, and are therefore more likely to achieve higher safety self-efficacy. On the contrary, workers with low proactive personalities tend to evaluate the environment negatively, generate psychological anxiety and burden, and passively accept risks in the working environment, thus decreasing their safety self-efficacy. There has been empirical research showing that a proactive personality makes a significant contribution to self-efficacy in a variety of different domains, such as job search self-efficacy [39] and career self-efficacy [40]. Therefore, the following hypotheses are proposed in this study.

**Hypothesis** **2:**
*Proactive personality is positively associated with safety self-efficacy.*


The important role of self-efficacy as an internal motivational factor that individuals exhibit, in terms of behavioral intentions and actual behavior, has been confirmed by numerous empirical studies [41,42]. A meta-analysis by Wang [43] showed that the correlations between self-efficacy and behavioral intentions, and actual behaviors were r = 0.63 and r = 0.46, respectively. In studies of safety behavior, higher safety self-efficacy can be translated into greater participation in safety activities. In fact, it has been recognized as one of the critical predictors of safety behavior [44]. Many empirical studies have highlighted the role of safety self-efficacy in promoting safety behaviors [44,45,46]. Specifically, workers with high safety self-efficacy have higher levels of confidence when faced with risk and are often motivated to apply effective ways to control their safety behaviors and deal with potential negative outcomes. This results in increased safety compliance and safety participation. On the contrary, if workers have low safety self-efficacy, their attitude in dealing with work tasks is more negative, while their perception of control over the results of their work is reduced, and they usually violate the rules and lack the awareness of proactive safety compliance. This leads to low levels of safety behaviors. Therefore, the following hypotheses are proposed in this study.

**Hypothesis** **3a:****:** *Safety self-efficacy is positively associated with safety compliance.*

**Hypothesis** **3b:**
*Safety self-efficacy is positively correlated with safety participation.*


Summarizing, the present study attempts to verify the mediating role of safety self-efficacy between proactive personality and safety behavior. This has not been verified in previous studies, particularly in the construction worker environment. Based on social cognitive theory, this study argues that a worker’s proactive personality first affects the degree of confidence and the way workers judge whether they are capable of completing safety-related tasks. Workers with high proactive personality are more confident in their abilities, have strong internal motivation, believe they are capable of carrying out their work safely, and form higher safety self-efficacy. At the same time, workers with high safety self-efficacy are not afraid of difficulties and risks at work, translate their abilities into work practices, control their behavior through effective ways, and deal with potential negative outcomes. This results in higher levels of safe behavior. Therefore, the following hypotheses are proposed in this study.

**Hypothesis** **4a:***Safety self-efficacy mediates the relationship between proactive personality and safety compliance*.

**Hypothesis** **4b:**
*Safety self-efficacy mediates the relationship between proactive personality and safety participation.*


### 2.3. The Mediating Role of Team Member Exchange

Based on social exchange theory, Graen and Cashman [47] first proposed the concept of leader–member exchange, which means that “in the workplace, workers’ behavior was influenced by the leader and other members”. Subsequently, Seers [24] argued that in the process of building construction activities, there exists not only a vertical relationship of leadership influencing employees’ behavior, but also a broad horizontal relationship of mutual influence between employees and staff. Therefore, Seers [25] proposed the concept of team member exchange, which is defined as “the perception of an individual member of the overall exchange relationship between him and other members of the team”. At the same time, Seers [25] showed that team member exchange is also a type of social exchange relationship. This is distinct from team member exchange, but both of them can influence employees’ behavior and performance.

Recent research has found a significant positive relationship between proactive personality and leadership member exchange [12,48]. However, the building operations were mostly subcontracted and workers were usually employed by small subcontractors and maintained a separated relationship with the upper levels of the project leaders [49]. This indicates that their behavior was not always effectively controlled under the formal rules and regulations established by the owner or general contractor, making the behavior of the workers largely beyond the control of the formal organization [5], thus leading to the existence of more employee-to-employee communication. In particular, the work tasks involving safety hazards rely on the tacit cooperation and mutual communication between employees to complete. With the absence of leaders, or when leaders provide insufficient resources, workers usually spend most of their time cooperating with each other and see their relationships with colleagues as a valuable resource because good co-worker relationships (i.e., high team member exchange) can provide help and assistance to help them accomplish their tasks [50]. Therefore, this study suggests that proactive personality may promote higher team member exchange. Because workers with a high proactive personality take action to change the environment that affects them, this has the potential to include the communication and exchange environment between them and other workers. Specifically, proactive workers may take the first step to offer to assist or help their colleagues, whether they ask for help or not. Because proactive workers have a high level of commitment to organizational goals, co-workers are more likely to offer help in a reciprocal manner [51]. Therefore, workers with higher levels of a proactive personality also have a higher quality of exchange relationships with their colleagues. On the contrary, individuals with lower levels of proactive personality are relatively passive and lack the awareness of proactive communication and interaction with other employees [9]. Therefore, the following hypothesis is proposed in this study.

**Hypothesis** **5:**
*Proactive personality is positively correlated with team member exchange.*


From the viewpoint of social exchange theory and team member exchange as being critical factors influencing employee behavior, proactive personalities have been shown to have a significant impact on organizational and individual behavior and performance [25,52,53]. Concretely expressed, in the process of interacting with colleagues, these individuals are willing, and have more opportunities, to share their views with others and put forward constructive opinions and suggestions regarding their work. They have a positive impact on other colleagues around them, influencing their behavior and performance [54,55]. Effects on safety behavior is no exception, which, as mentioned in the introduction, has been conceptualized as a special kind of job performance behavior whereby workers actively comply with safety rules and regulations [13]. Applying these theories to building construction workers’ safety behavior, workers with high team member exchange are likely to receive more help, advice, and feedback from their colleagues than those with low team member exchange, and in order to maintain this high exchange relationship, they must provide their help when requested by their colleague [25,50]. Therefore, teams with high-quality team member exchanges are more likely to exhibit smooth and effective workflow [56], especially in the context of construction safety where high-quality team member exchanges are oriented toward high safety performance goals. The result is a series of shared work experiences and mutual assistance in solving work challenges which increases safety compliance and safety participation of workers. Conversely, low-quality team member exchanges tend to reduce inter-employee communication, resulting in less motivation to participate in safety behaviors. Therefore, the following hypotheses are proposed in this study.

**Hypothesis** **6a:**
*Team member exchange is positively correlated with safety compliance.*


**Hypothesis** **6b:**
*Team member exchange is positively correlated with safety participation.*


Summarizing, social exchange theory suggests that team member exchange should mediate between proactive personality and safe behavior. Workers with high proactive personality in good social relationships are able to actively communicate with their colleagues at work and are more sensitive to interpersonal interactions. In addition to work-based collaboration, there are other social life and emotional exchanges with team members, forming a high-quality team member exchange relationship. This can enable construction workers to perform with higher levels of safe behavior. On the contrary, workers with low proactive personality are susceptible to environmental factors, do not have a sense of teamwork, and do not easily form team member exchange relationships. This leads to lower levels of lower safety behaviors. Therefore, the following hypotheses are proposed in this study.

**Hypothesis** **7a:**
*Team member exchange mediates between proactive personality and safety compliance.*


**Hypothesis** **7b:**
*Team member exchange mediates between proactive personality and*
*safety participation.*


### 2.4. The Moderating Role of Safety-Specific Transformational Leadership

Trait activation theory combines traits and situations, emphasizing that personality traits require relevant situations to express trait-related behaviors [16]. In other words, the expression of trait behavior depends on the trait-related contexts. It has been shown that leadership style not only directly influences workers’ behavior, but also plays a moderating role in the influence of personality traits on workers’ behavior [5,57,58].

Transformational leadership is a trait-related characteristic that induces and guides proactive workers to exhibit relevant behaviors [9]. However, in the field of occupational safety, there are few studies on the moderators of proactive personality influencing safety behavior. Therefore, in this study, we predicted that safety-specific transformational leadership is a contextual factor that enhances proactive personality in relation to safety behaviors. Safety-specific transformational leadership is defined as “the act of providing workers with a shared vision of safety and the motivation, skills, and self-efficacy needed to achieve that vision [27]. When proactive workers work with safety-specific transformational leaders, their safety behaviors may be guided by these leadership behaviors and they ultimately exhibit higher levels of safety behaviors themselves. In other words, when supervisors demonstrate higher levels of safety-specific transformational leadership, they send clear signals to construction workers that proactive personalities are supported and safe behaviors are expected, desired, supported, and rewarded, creating a strongly safety-oriented environment. Therefore, the following hypotheses are proposed in this study.

**Hypothesis** **8a:**
*Safety-specific transformational leadership will moderate the positive relationship between proactive personality and safety compliance. This relationship is more pronounced when safety-specific transformational leadership is high rather than when it is low.*


**Hypothesis** **8b:**
*Safety-specific transformational leadership will moderate the positive relationship between proactive personality and safety participation. This relationship is more pronounced when safety-specific transformational leadership is high rather than when it is low.*


## 3. Methods

### 3.1. Sample and Procedure

To recruit the participants, we followed the standard procedures suggested in previous research on construction worker’s safety behavior [59,60]. A phased approach to data collection was adopted to avoid the influences of common method variance. Before collecting data, each of the participants was informed of the purpose and procedures of the research, and their written consent was obtained. Confidentiality of their information was guaranteed. Additionally, during the questionnaire process, each participant received a unique code for survey matching. This was removed from the survey after all the responses had been collected.

The survey questionnaires collected data from 10 Chinese construction sites covering transportation, bridge building, and housing. In order to ensure accuracy and achieve a high recovery rate, data were collected using offline questionnaires. These were answered by the employees in their spare time after work. Any questions that the employees themselves felt were more ambiguous were explained by members of the survey research team. The survey was divided into two phases. In the first phase, researchers collected information on the evaluation of demographic variables, proactive personality, team member exchange, and safety self-efficacy. Two months later (to avoid respondent habitual thinking causing invalidity to collected data), a second survey was conducted with the respondents from the first survey as the subjects. In the second phase, researchers collected information on the evaluation of safety-specific transformational leadership and safety behavior (safety compliance and safety participation). Finally, we matched and integrated the two surveys (such as matching Zhang San 1 with Zhang San 2, Li Si 1 with Li Si 2) and then eliminated invalid questionnaires.

Out of 368 questionnaires, a total of 287 were valid (a 78% return rate) after excluding the incomplete returns. This is in line with the sample size required for this type of study [61]. Of these samples, 85.7% (*n* = 246) of the workers were male and 14.3% (*n* = 41) were female. In terms of age, 14.6% (*n* = 42) of the workers were between 20 and 29 years old, 31.7% (*n* = 91) were between 30 and 39 years old, 34.1% (*n* = 98) were between 40 and 49 years old, and 19.5% (*n* = 56) were 50 years old and above. In terms of work experience, 36.6% (*n* = 105) of the workers had 0–5 years of experience. In addition, 27.5% (*n* = 79) of the workers had 6–10 years of experience and 35.9% (*n* = 103) of the workers had 11 years of experience and above.

### 3.2. Measures

Questionnaire items were marked using a seven-point Likert scale measuring from 1 = *absolutely disagree* to 7 = *absolutely agree*. The scales were translated from English to Chinese through translation and back translation procedures. Measurement of Proactive personality and other variables are described below:

Proactive personality: Proactive personality was measured using a 10-item scale developed by Seibert [62]. One example item is “*Wherever I have been, I have been a powerful source for constructive change*”. The Cronbach’s coefficient of the proactive personality in this was 0.949. Standardized factor loading values were in the range of 0.758 to 0.825.

Safety self-efficacy: The 6-item general self-efficacy scale developed by Chen [63] was used and each item was transformed to reflect safety self-efficacy for this study. As previously mentioned, specific self-efficacy is more significant in shaping individual-specific behaviors in terms of facilitation, so we constructed the safety self-efficacy scale by focusing the questions of the general self-efficacy scale on the safety domain. Sample items include: “*I can successfully overcome many safety -related challenges*” and “*I have confidence that I can efficiently handle various safety-related tasks*”. The Cronbach’s coefficient of the safety self-efficacy in this study was 0.915. Standardized factor loading values were in the range of 0.739 to 0.834.

Team member exchange: The 10-item scale developed by Seers [24] was used to measure team member exchange. Two examples of team member exchange items are: “*I was willing to help team members with their work assignments*” and “*I frequently recommend better ways of working for other team members*”. The Cronbach’s coefficient of the team member exchange in this study was 0.922. Standardized factor loading values were in the range of 0.668 to 0.753.

Safety-specific transformational leadership: A 9-item scale developed by Barling [64] was applied to measure safety-specific transformational leadership. An example of a safety-specific transformational leadership item is: “*supervisor explained to us the vital elements of maintaining the safe working environment*”. The Cronbach’s coefficient of the safety-specific transformational leadership in this study was 0.929. Standardized factor loading values were in the range of 0.696 to 0.833.

Safety behavior: Safety behaviors were measured using a 7-item scale developed by Griffin and Neal [13], which has been extensively applied in the construction industry [5,65] and included for safety compliance and safety participation dimensions. An example of a safety participation item is: “*I voluntarily carry out tasks or activities that help improve work place safety*”. An example of a safety compliance item is: “*I use all necessary safety equipment to do my job*”. The Cronbach’s coefficients of safety compliance and safety participation in this study were 0.826 and 0.905. Standardized factor loading values were in the range of 0.812 to 0.841.

Control Variables: Recent studies have demonstrated that certain demographic factors such as gender, age, and work experience can dramatically influence the safety behavior of construction workers [65,66]. Hence, gender, age, and work experience were used as control variables in this study.

## 4. Results

### 4.1. Statistical Analysis of Data

SPSS 21.0, AMOS 21.0 and PROCES [67] were used for statistical analysis of the data collected from this survey. First step: confirmatory factor analysis was used to test the issue of discriminant validity and possible common method bias among the key variables involved in this study. Second step: descriptive statistics and correlation analysis were conducted for each study variable, while convergent validity and composite reliability were tested. Third step: the theoretical hypotheses were tested using hierarchical regression.

### 4.2. Confirmatory Factor Analysis and Homologous Deviation Analysis

Confirmatory factor analysis was used to test the fit of the measurement’s models and to select the model with the best fit by comparing the models. Table 1 shows the fit indices of all tested models. Comparison of the results revealed that the six factor model fit was better (χ2 = 1169.326, df = 804, χ2/df = 1.454 < 3, RMSEA = 0.040 < 0.05, CFI = 0.955 > 0.9, TLI = 0.952 > 0.9, SRMR = 0.044 < 0.05) and the fit was significantly better than the other models, indicating that the measurement scale used in this study had excellent discriminant validity [68].

Since the survey data were collected by self-assessment and each questionnaire involved key variables provided by the same participant, there was the possibility of common method bias. Although anonymous measurements and multiple data sources were used to reduce the effect of common method bias, there was still a need for a common method bias test on the measurement data. In this study, the common method bias was tested using the “confirmatory factor analysis that adds common method factors” method recommended by Xiong [69].

As survey data were collected through self-evaluation, with each questionnaire involving key variables provided by the same subject, there may be a possibility of common method bias. To minimize this bias, this study employed anonymous measurement, phased and multi-source questionnaire methods. Additionally, SPSS 22.0 software was used to conduct Harman’s single factor test and factor analysis on all variables to analyze the collected data. The results indicated that there were six factors with eigenvalues greater than one (more than one), and the largest factor variance explained rate was 28.97% (less than 40%). Therefore, there was no significant common method bias [69,70].

### 4.3. Means, Standard Deviations, and Correlations

Table 2 shows descriptive statistics: Pearson’s correlation coefficient, average variance extracted (AVE), and construct reliability (CR). First, for the correlation analysis, proactive personality was positively correlated with safety self-efficacy (r = 0.396, *p* < 0.01), team member exchange (r = 0.337, *p* < 0.01), safety compliance (r = 0.367, *p* < 0.01), safety participation (r = 0.334, *p* < 0.01), and safety-specific transformational leadership (r = 0.143, *p* < 0.05). Safety self-efficacy was positively related to safety compliance (r = 0.368, *p* < 0.01) and safety participation (r = 0.299, *p* < 0.01), as was team member exchange with safety compliance (r = 0.317, *p* < 0.01) and safety participation (r = 0.341, *p* < 0.01). Secondly, the average variance extracted (AVE) values for each construct were all greater than 0.5 and the construct reliability (CR) was greater than 0.8 [71], indicating good convergent validity of the structural model in this study.

### 4.4. Hypotheses Testing

First, the mediating effects of safety self-efficacy and team member exchange in the relationship between proactive personality and safety behavior were examined using Model 4 (Model 4 is a simple mediation model) in the process program in SPSS prepared by Hayes [67], controlling for gender, age, and work experience. The results (Table 3 and Table 4) show that proactive personality had a significant positive effect on safety compliance (*B* = 0.135, *t* = 3.363, *p* <0.001) and safety participation (*B* = 0.126, *t* = 3.069, *p* < 0.001), and when mediating variables were put in, proactive personality had a significant positive effect on safety compliance (*B* = 0.135, *t* = 3.363, *p* < 0.001), and safety participation (*B* = 0.126, *t* = 3.069, *p* < 0.001) remained significant direct predictors of, therefore, hypotheses H1a and H1b are supported. Proactive personality had a significant positive effect on safety self-efficacy (*B* = 0.279, *t* = 7.034, *p* < 0.001) and team member exchange (*B* = 0.234, *t* = 5.930, *p* < 0.001), therefore, hypotheses H2 and H5 were supported. Safety self-efficacy had a significant positive effect on safety participation (*B* = 0.131, *t* = 2301, *p* < 0. 001) and safety compliance (*B* = 0.206, *t* = 3.719, *p* < 0.001), therefore, hypotheses H3a and H3b were supported. Team member exchange had a significant positive effect on safety participation (*B* = 0.212, *t* = 3.706, *p* < 0.001) and safety compliance (*B* = 0.158, *t* = 2.838, *p* < 0.001), therefore, hypotheses H6a and H6b were supported. Furthermore, the direct effect of proactive personality on safety compliance and safety participation, and the mediating effect of safety self-efficacy and team member exchange, did not contained 0 at the upper and lower limits of the bootstrap 95% confidence interval (Table 4), indicating that proactive personality not only predicted safety behavior directly, but also could predict safety behavior (safety compliance and safety participation) through the mediating effect of safety self-efficacy and team member exchange, therefore, hypotheses H4a, H4b, H7a, and H7b were supported. Moreover, in the direct and mediating effects models, proactive personality explained 16.1% (F = 13.564, *p* < 0.001) of the variance in safety self-efficacy, 12.2% (F = 9.760, *p* < 0.001) of the variance in team member exchange, 26.1% (F = 16.469, *p* < 0.001) of the variance in safety compliance behavior, and 21.7% (F = 12.950, *p* < 0.001) of the variance in safety participation behavior. In the moderating effects model, proactive personality explained 27.5% (F = 13.146, *p* < 0.001) of the variance in safety compliance behavior and 30.6% (F = 15.325, *p* < 0.001) of the variance in safety participation behavior.

Second, the moderating effect of safety-specific transformational leadership between proactive personality and safety behaviors (safety compliance and safety participation) was tested using Model 5 (Model 5 is a moderating model of direct effects, which is consistent with the theoretical model of this study) in the PROCESS program in SPSS prepared by Hayes [67]. The results (Table 3) show that when safety-specific transformational leadership is put into the model, the product term of proactive personality and safety-specific transformational leadership has a significant predictive effect on safety compliance (*B* = 0.142, *t* = 3.417, *p* < 0.001) and safety participation (*B* = 0.111, *t* = 2.592, *p* < 0.01), indicating that safety-specific transformational leadership could play a moderating effect in the direct prediction of proactive personality on safety compliance and safety participation, therefore, hypotheses H8a and H8b were supported.

Further simple slope analysis, as shown in Figure 2, for subjects with low levels of safety-specific transformational leadership (M − 1SD), proactive personality did not have a significant effect on safety compliance behavior (simple slope = 0.020, *t* = 0.316, *p* > 0.05), while for subjects with high levels of safety-specific transformational leadership (M + 1SD), proactive personality had a positive predictive effect on safety compliance behavior (simple slope = 0.242, *t* = 3.826, *p* < 0.001), indicating that as the level of safety-specific transformational leadership increased, the predictive effect of proactive personality on safety compliance behavior tended to increase gradually (Table 5). As shown in Figure 3, for subjects with low levels of safety-specific transformational leadership (M − 1SD), proactive personality did not have a significant effect on safety participation (simple slope = 0. 011, t = 0.174, *p* > 0.05), while for subjects with high levels of safety-specific transformational leadership (M + 1SD), proactive personality had a positive predictive effect on safety participation (simple slope = 0.233, *t* = 3.684, *p* < 0.001), indicating that as the level of safety-specific transformational leadership increased, the predictive effect of proactive personality on safety participation tended to increase gradually (Table 5). In addition, the moderating effect of safety-specific transformational leadership in proactive personality and safety compliance was greater than that of proactive personality and safety participation.

## 5. Discussion

Based on social cognitive theory, social exchange theory, and trait activation theory, this study proposes and tests a dual mediating role and a moderating mechanism of action for the influence of proactive personality on safety behavior in the context of construction safety. Not only is the question of how proactive personality affects workers’ safety behavior (through the mediating role of safety self-efficacy and team member exchange) clarified, but also the question of under what conditions proactive personality has a more significant impact on workers’ safety behavior is addressed (through the moderating role of safety-specific transformational leadership). The findings have theoretical and practical implications for deepening research on the relationship between proactive personality and safety behavior and for guiding managers and workers on how to improve safety behavior.

### 5.1. Theoretical Implications

This study has three major theoretical contributions to the literature.

First, it enriches the research on the correlation between personality traits and workers’ safety behavior. The important role of proactive personality in promoting safe behavior among construction workers is revealed in the context of safety. In prior studies, most scholars have researched the influence of the Big Five personality traits on safety behavior [1,21,34] and the present study supplements this research by determining the influence of personality traits on safety behavior by demonstrating the influence of proactive personality on safety behavior. However, as for proactive personality, as a unique personality trait not covered in the Big Five [35], it is very important for workers to have a high proactive personality, especially in such construction activities where there are higher risks, complex environments, and higher mobility [7]. As previous studies have pointed out, workers with proactive personalities are not constrained by their environment at work and have a tendency to take proactive action to change the external environment in order to achieve their goals [6]. Therefore, it is important to focus on the proactive personality of workers in the context of workplace safety as they will aim to improve their safety participation and safety compliance behavior.

Second, drawing on social cognitive theory and social exchange theory, this study constructs a more comprehensive theoretical framework of safety behavior by considering safety self-efficacy and team member exchange as mediating variables between proactive personality and safety behavior. The findings covering these pathways fill a gap in the literature and are the first attempt to open the “black box” between proactive personality and safety behavior. To begin with, we responded to the call of Xia [5], that personality traits have the potential to influence safety behavior through motivation. Self-efficacy is a motivational factor, and the higher the employee’s self-efficacy, the stronger will be the internal motivation [72]. However, past research has also shown that self-efficacy can be used as a mediating variable to influence the relationship between proactive personality and behavior or performance [8,22]. As expected, the present study found that safety self-efficacy supports a mediating role between proactive personality and safety behavior in a safety context, a finding consistent with Tierney’s [37] findings that domain-specific, task-specific, and problem-specific self-efficacy is more predictive of behavior or performance than general self-efficacy. In addition, our empirical findings support Li’s [44] claim that safety self-efficacy is decisive in predicting safety behaviors. Therefore, we should pay attention to and promote workers’ safety self-efficacy. Next, we responded to Liu’s [73] call to investigate the mediating role of team member exchange in proactive personality and safety behavior in this benign exchange relationship. Seers [24] highlights that most previous research has focused on the effects of leader–member exchange on employee behavior and performance, ignoring the employee–employee exchange relationship. Subsequently, Seers [25] argues that team member exchange, as an extension of social exchange relationships, can predict employee behavior and performance as well as leader member exchange. As expected, our study found that team member exchange similarly supported a mediating role in proactive personality and safety behavior. As previously explained, building construction is mostly subcontracted, and workers are usually employed by small subcontractors and maintain a separate relationship with the upper levels of the project [49]. This suggests that their behavior is not always effectively controlled by the formal rules and regulations established by the owner or general contractor, making the behavior of the workers largely beyond the control of the formal organization. This leads to the existence of more employee-to-employee communication. Our results suggest that if proactive workers establish a good social exchange relationship with their co-workers, this relationship creates a reciprocal behavior that exhibits as higher safety participation and safety compliance behavior. Therefore, we should pay attention to these team member exchange relationships among workers and consider this as an important variable in shaping employee safety behavior in future studies. Furthermore, we found that the mediating effect of safety self-efficacy accounts for a higher proportion of the effect of proactive personality on safety compliance behavior than that of team member exchange. In contrast, the mediating effect of team member exchange accounts for a higher proportion of the effect of proactive personality on safety participation behavior than that of safety self-efficacy. This is because safety compliance behavior is an internal safety behavior that is largely influenced by individual factors, and safety self-efficacy is an individual factor that predicts internal safety behavior. On the other hand, safety participation behavior is an external safety behavior that is largely influenced by organizational factors, and team member exchange is an organizational factor that predicts external safety behavior.

Third, considering the external conditions that promote proactive workers to exhibit more safe behaviors, trait activation theory suggests, in particular, that the effects of traits are more likely to be elicited by appropriate external situational stimuli under adverse circumstances [16]. However, as Grant [26] pointed out, motivated workers do not always exhibit behavior that is beneficial to the organization, and recent research has demonstrated a moderating effect of situational factors such as leadership style on the relationship between personality traits and behavior [5,74]. As anticipated, this study also found a moderating effect: that of safety-specific transformational leadership in the context of safety in proactive personality and safety behaviors. This is different from the results of Lai’s [9] study which found no significant moderating effect of transformational leadership on proactive personality and proactive behavior with corporate employees. However, the reason for the difference is that nearly 83% of the Lai study employee participants had been in their current positions for at least 1 year. These employees were therefore likely to be familiar with their leader’s leadership style and understand which behaviors are encouraged and which are prohibited. Therefore, there is no moderating effect of transformational leadership on employees’ proactive influence. In contrast, as stated earlier, the construction workforce has a more mobile and temporary nature, working on individual projects for fairly limited periods of time, being unfamiliar with safety regulations and project rules and behaving in ways that are largely beyond the scope of formal organizational control. When increasing the interaction between safety transformational leaders and workers, workers can be facilitated to exhibit higher levels of safety behaviors. Therefore, our findings are justified and we should focus on employee proactive personality traits along with safety transformational leaders’ guidance of workers especially in more transient work environments.

### 5.2. Practical Implications

This study provides some practical implications for improving the safety behavior of construction workers.

First, the results of the study emphasize the important role of proactive personality in construction workers’ safety behaviors. Therefore, assessing the proactive personality of workers in the staff recruitment process and scientifically examining the differences in personality traits and other aspects of employees can be done to proactively reduce the possible unsafe behaviors beforehand. In particular, for special operators, it is important not only to strictly ensure that they are licensed to work, but also to hire workers with high proactive personalities. In addition, when training workers and adjusting positions, we should also pay attention to the assessment of the personality traits of employees and select, assign, and train workers according to the assessment results and job requirements, so as to better match people with jobs. Thus, you can improve the safety behavior of workers and the safety performance of enterprises.

Second, the results also highlight the mediating role of safety self-efficacy and team member exchange in proactive personality and safety behavior. To begin with, enhancing workers’ safety self-efficacy is as important to employees’ safety behavior as self-confidence is to individual behavior motivation. This is because safety self-efficacy, like internal motivation, plays a decisive role in predicting safety behavior [44]. In practice, managers should take care of their workers’ working conditions and actively support their personalized care to ensure that they are energetic and can concentrate on their work. In addition, workers should be encouraged to participate in safety-related meetings, given the opportunity to express their ideas and suggestions, and their reasonable, relevant suggestions and solutions should be adopted and rewarded as much as possible, thus motivating employees to develop their own abilities. In this way, workers are: (i) more likely to improve their confidence and motivation to complete safety-related work tasks (i.e.,; safety self-efficacy) so that they always maintain a higher high safety self-efficacy, (ii) more willing to take the initiative to comply with safety rules and regulations, and (iii), likely to obey safety management and actively participate in safety promotion, thus improving their own safety behavior. With team work becoming common, workers are increasingly practicing getting along and interacting with colleagues in the workplace, and so the quality of team member exchange relationships has a significant impact on employee behavior and performance [9,25]. In particular, in most building construction activity, more work tasks are accomplished through mutual support and collaboration between teams. However, in situations where the leadership provides limited resources, managers should pay attention to the improvement of the quality of team members’ exchange, create opportunities for team members to get to know each other, and get familiar and accept each other as much as possible. For example, an organization could hold staff construction safety exchange meetings and routinely conduct construction safety training from time to time, solve safety-related problems for employees in a way that provides guidance, support employees’ ability to express themselves and to identify problems and solve them themselves, form a good working environment for safety, and promote the formation of mutual assistance and reciprocity among team members.

Third, our findings suggest that the interaction of safety-specific transformational leadership and proactive personality affects workers’ safety behaviors. Therefore, in practice, we should not only focus on employee safety training, but also on selecting and training leaders according to the characteristics of safety-specific transformational leadership. In building construction activities, safety-specific transformational leaders lead by example, teach by example, and guide workers with proactive personalities to exhibit higher levels of safety participation and safety compliance behaviors.

## 6. Limitations and Future Research

Although every effort was made to ensure the objectivity and scientific validity of the study, there were some limitations. First, due to the constraints of research cost and time and other conditions, a cross-sectional survey study was used, and there may be some bias in the survey data as well as insufficient analysis of the influence of persuasive power of proactive personality on safety behavior. Therefore, in future studies, we will try to expand the sample size and use the survey method in a longitudinal study to ensure the generalizability of the research results. Second, the current study only controlled for gender, age, and work experience [60,65,66], which are commonly used control variables in safety behavior research. However, there are other variables related to safety behaviors that have not been considered for control, such as the Big Five personality traits [1,21,34]. Therefore, further research is needed in future studies to incorporate more control variables and to test the studied model more conservatively.

## 7. Conclusions

This study relies on social cognitive theory, social exchange theory, and trait activation theory to better understand the mechanism by which having proactive personality traits influences construction workers’ safety behavior. First, the study demonstrates that proactive personality not only directly has a significant positive effect on construction workers’ safety behavior, but at the same time, also indirectly influences this behavior through safety self-efficacy and team member exchange. The mediating effect of safety self-efficacy accounts for a higher proportion of the effect of proactive personality on safety compliance behavior than that of team member exchange. In contrast, the mediating effect of team member exchange accounts for a higher proportion of the effect of proactive personality on safety participation behavior than that of safety self-efficacy. Second, the study also demonstrates that safety-specific transformational leadership enhanced the effect of proactive personality on safety behaviors. Therefore, these findings suggest that in order to improve the level of safety behavior among construction workers, it is important to examine the level of workers’ proactive personality while also developing a group of safety-specific transformational leaders. These safety-specific transformational leaders can be considered as an intervention to improve construction workers’ safety behavior.

## Figures and Tables

**Figure 1 behavsci-13-00337-f001:**
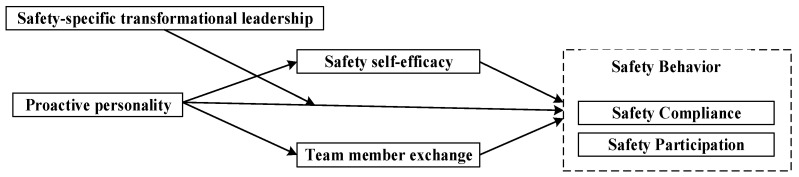
Model hypothesis.

**Figure 2 behavsci-13-00337-f002:**
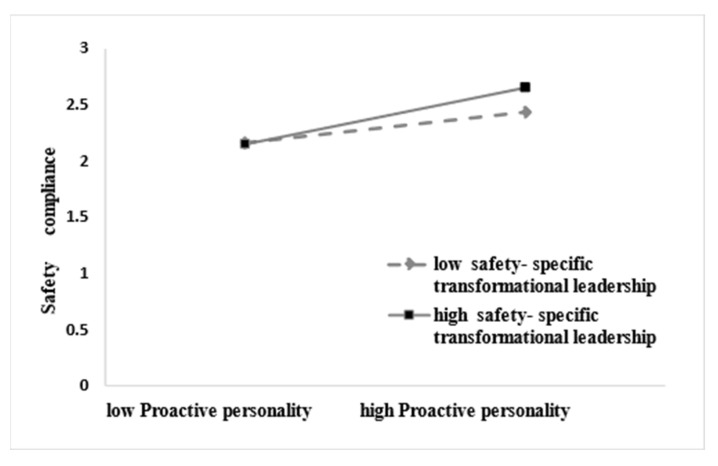
Interactive effect of proactive personality and safety-specific transformational leadership on safety compliance.

**Figure 3 behavsci-13-00337-f003:**
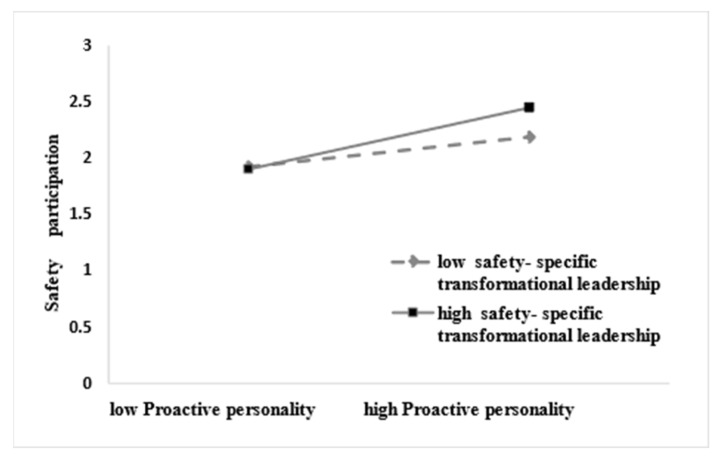
Interactive effect of proactive personality and safety-specific transformational leadership on safety participation.

**Table 1 behavsci-13-00337-t001:** Fit indices for the measurement models.

Model	χ2	df	χ2/df	CFI	TLI	RMSEA	SRMR
Hypothesized six-factor model	1169.326	804	1.454	0.955	0.952	0.040	0.044
Five-factor model 1	2793.885	809	3.453	0.755	0.739	0.093	0.138
Five-factor model 2	2387.384	809	2.951	0.805	0.793	0.083	0.114
Four-factor model 1	3689.331	813	4.538	0.645	0.624	0.111	0.154
Four-factor model 2	4045.068	813	4.975	0.601	0.577	0.118	0.160
Three-factor model	4897.107	816	6.001	0.496	0.468	0.132	0.169
Two-factor model	5233.558	818	6.398	0.455	0.426	0.137	0.175
One-factor model	5793.298	819	7.074	0.386	0.354	0.146	0.171

Note: χ2 = chi-square; df = degree of freedom; five-factor model1 = proactive personality and safety-specific transformational leadership combined; five-factor model2 = team member exchange and safety-specific transformational leadership combined; four-factor model1 = proactive personality, safety self-efficacy, and safety-specific transformational leadership combined; four-factor model2 = proactive personality, team member exchange, and safety-specific transformational leadership combined; three-factor model = proactive personality, safety self-efficacy, team member exchange, and safety-specific transformational leadership combined; and two-factor model = safety compliance and safety participation combined, proactive personality, safety self-efficacy, team member exchange, and safety-specific transformational leadership combined.

**Table 2 behavsci-13-00337-t002:** Descriptive statistics, Pearson’s correlation coefficients, average variance extracted (AVE), and construct reliability (CR).

Variable	1	2	3	4	5	4	7	8	9
1. gender	N/A								
2. age	−0.031	N/A							
3. work experience	−0.067	0.758 **	N/A						
4. Proactive personality	−0.050	0.159 **	0.134 *	**(0.648)**					
5. Safety self-efficacy	−0.017	0.103	0.117 *	0.396 **	**(0.630)**				
6. Team member exchange	0.064	0.086	0.069	0.337 **	0.331 **	**(0.543)**			
7. Safety compliance	0.021	0.275 **	0.239 **	0.367 **	0.368 **	0.317 **	**(0.710)**		
8. Safety participation	0.087	0.255 **	0.240 **	0.334 **	0.299 **	0.341 **	0.390 **	**(0.678)**	
9. Safety-specific transformational leadership	−0.005	0.052	0.071	0.143 *	0.034	0.372 **	0.224 **	0.253 **	**(0.594)**
Mean	1.143	2.585	1.993	3.822	4.191	4.132	4.031	3.993	3.447
SD	0.351	0.964	0.853	1.196	0.856	0.833	0.824	0.832	0.851
CR				0.949	0.898	0.922	0.905	0.863	0.929

Note: * *p* < 0.05, ** *p* < 0.01, and *** *p* < 0.001; N/A = not applicable; SD = standard deviation; and the values inside the brackets on the diagonal represent AVE.

**Table 3 behavsci-13-00337-t003:** Hierarchical regression analysis of mediation and moderation effects.

Variables	Model4								Model5			
	Direct and Mediating Effects								Moderating Effects			
	Safety Self-Efficacy		TME		Safety Participation		Safety Compliance		Safety Participation		Safety Compliance	
	*B*	*t*	*B*	*t*	*B*	*t*	*B*	*t*	*B*	*t*	*B*	*t*
gender	0.017	0.123	0.196	1.473	0.227	1.817	0.077	0.630	0.109	0.920	0.255	2.085 *
age	−0.017	−0.225	0.027	0.364	0.106	1.523	0.148	2.1852 *	0.121	1.825	0.086	1.261
work experience	0.080	0.947	0.006	0.069	0.096	1.228	0.046	0.605	0.044	0.593	0.091	1.183
Proactive personality	0.279	7.034 ***	0.234	5.930 ***	0.126	3.069 ***	0.135	3.363 ***	0.131	3.367 ***	0.122	3.023 **
Safety self-efficacy					0.131	2.301 ***	0.206	3.719 ***	0.222	4.077 ***	0.149	2.650 **
TME					0.212	3.706 ***	0.158	2.838 ***	0.122	2.090 *	0.165	2.731 **
STL									0.114	2.1482 *	0.136	2.491 *
Interactive									0.142	3.417 ***	0.111	2.592 **
R	0.402		0.349		0.466		0.511		0.553		0.524	
R^2^	0.161		0.122		0.217		0.261		0.306		0.275	
F	13.564 ***		9.760 ***		12.950 ***		16.469 ***		15.325 ***		13.146 ***	

Note: * *p* < 0.05, ** *p* < 0.01, and *** *p* < 0.001; N/A = not applicable; TME = Team member exchange; STL = Safety-specific transformational leadership; Interactive = Proactive personality X Safety-specific transformational leadership; mediation and moderation effects were tested using bootstrapping analysis [5000 repetitions, 95% confidence intervals (CI)].

**Table 4 behavsci-13-00337-t004:** Decomposition of total effect, direct effect, and mediating effect.

Dependent Variable		Effect	SE	LL 95% CI	UL 95% CI	Effect Ratio
Safety participation	Total effect	0.212	0.040	0.141	0.295	
	Direct effect	0.126	0.046	0.039	0.220	0.594
	Mediating effects of safety self-efficacy	0.037	0.018	0.005	0.074	0.172
	Mediation effect of team member exchange	0.050	0.017	0.020	0.086	0.233
Safety compliance	Total effect	0.229	0.039	0.156	0.310	
	Direct effect	0.135	0.046	0.047	0.223	0.588
	Mediating effects of safety self-efficacy	0.058	0.016	0.028	0.091	0.251
	Mediation effect of team member exchange	0.037	0.016	0.009	0.073	0.161

Note: Indirect effects were tested using bootstrapping analysis [5000 repetitions, 95% confidence intervals (CI)]; LL = lower level; and UL = upper level.

**Table 5 behavsci-13-00337-t005:** Direct effects of different levels of safety-specific transformational leadership.

	Safety-Specific Transformational Leadership	Effect	SE	LL 95% CI	UL 95% CI
Proactive personality → safety compliance	−1.023 (M − 1SD)	0.019	0.052	−0.919	0.114
	(M)	0.131	0.039	0.061	0.214
	1.023 (M + 1SD)	0.252	0.053	0.148	0.356
Proactive personality → safety participation	−0.851 (M − 1SD)	0.027	0.054	−0.079	0.134
	(M)	0.122	0.040	0.043	0.201
	0.851 (M + 1SD)	0.216	0.055	0.109	0.324

Note: Moderation effects were tested using bootstrapping analysis [5000 repetitions, 95% confidence intervals (CI)]; LL = lower level; UL = upper level; M = Mean; and SD = standard deviation.

## Data Availability

All data, models, or code generated or used during the study are available from the corresponding author by request.

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
