# Peer review of "Proactive Personality and Construction Worker Safety Behavior: Safety Self-Efficacy and Team Member Exchange as Mediators and Safety-Specific Transformational Leadership as Moderators"

_behavsci, 2023, doi:10.3390/bs13040337_

Round 1
Reviewer 1 Report
The topic of this manuscript falls within the scope of Behavioral Sciences journal. The cross-sectional survey and consequent statistical analysis are well documented and their limitations identified. Theoretical background is based on substantial scientific literature, and the paper is overall well written.
The following comments refer mainly to editing corrections which should be included in the final proof.
Page 1, line 35. The term “production safety accidents” is not very common thus it is not clear if it is referring to construction accidents or all types of accidents, and how it is distinguished from “fatalities in housing and municipal construction”. This should be clarified and the terms production safety accidents replaced throughout the manuscript.
Page 2, line 67. Correct “ theories”
Page 2, lines74-76. Revise the phrase “therefore, many scholars…of such behavior”
Page 3, par.2.1 first paragraph, 7th line (no line-numbering?). Correct “introduced”
Page 4, par.2.2 Hypothesis 3b. Correct “was” to “is”
Page 5, par.2.2. Why the numbering of hypotheses is not continuous (hypotheses 6a & 6b follow hypotheses 3a & 3b)
Page 5, par.2.3 second paragraph, 2nd line. Correct “exchange”
Page 6, par.2.3. Hypothesis 5a is written twic3 (only the numbering)
Page 7, par.3.1 second paragraph, 8th line. Clarify the meaning of the parenthesis
Page 13, par.5.1 second paragraph. A word is missing after reference Tierney’s [37]
Page 13, par.5.1 last paragraph. Clarify the first sentence.
Page 14, par.5.2. This paragraph needs further review and editing, because in various parts there are spelling mistakes and unclear meanings.
Page 15, Conclusions. Correct last phase.
Reviewer 2 Report
The manuscript was reviewed. Overall, the article is interesting, conceived and presented well, and merits publication. However, a few changes are being recommended.
Attached is a pdf document which presents this reviewer’s comments in “red.” Many of the changes related to grammar. In a few places, the reviewer has also highlighted in yellow certain phrases/sentences that require discussion (which are discussed below).
- The sequence of hypotheses numbering was incorrect. The reviewer attempted to correct these mistakes throughout the document, but the authors need to recheck.
- The authors state that the two phases of the study were integrated. In order to make the study work, the respondents in the first survey needed to be connected somehow with their responses in the second survey. A few sentences describing how this was accomplished should be presented in the manuscript. If there is not 100% assurance that the responses per individual were linked accurately from first to second survey, then the study is invalid.
- In the sample questions provided on self-efficacy, the questions as they are presented deal with security and not safety. Safety and security are related but not identical concepts. Are you sure the questions were translated correctly into English? If it was translated correctly, then the authors need to make the case that safety and security are equivalent concepts for the purpose of their study.
- In the manuscript, the authors refer to confirmatory factor analysis models M1 and M2. The authors need to more specifically describe these models.
- The authors conducted hypothesis testing on the meditated model – Model 4 (M4?). What exactly is Model 4 and why is it called Model 4 when there does not seem to be a Model 3? – Perhaps showing it diagrammatically in a figure form makes sense. Also eventually showing your final mediated and moderated model results in figure form indicating the strength of direct and indirect relationships (often shown when researchers conduct structural equation modeling) would be a nice addition to your manuscript.
- I believe there were some errors in your Section 4.4 – Hypothesis testing when you presented some of your data. Check to see if I am correct in my rewrite.
- In Table 3, there seems to be a significant gender difference in safety compliance responses. This might be worthy of discussion.
- In Table 3, some commentary on the strength of the various models (either by discussing the F statistics or the R2 numbers) would be appropriate.
- In the conclusion section of the manuscript, the authors write: The mediating effect of safety self-efficacy accounts for a higher proportion of the effect of proactive personality on safety compliance behavior than that of team member exchange. In contrast, the mediating effect of team member exchange accounts for a higher proportion of the effect of proactive personality on safety participation behavior than that of safety self-efficacy. However, these findings are never really discussed in the body of the manuscript. When the authors present Table 4 in the manuscript, they should discuss these findings here. Also, the information in Table 4 is rich for enhanced discussion beyond what is presented in this manuscript.
- In discussing Table 5, you may want to define in a legend what “1SD” means (1 standard deviation). Also explicitly discussing which confidence limit ranges for data sets do not pass through zero (and thus indicates a significant relationship) would be nice.
- The authors state that the results of this study reveal not only can proactive personality influence construction workers’ safety behaviors, but also that proactive personality influences safety behaviors to a greater extent than Big Five personality influence safety behaviors. This is a pretty “big” declaration which is not supported by any specific information in the manuscript. How did the authors reach this conclusion?
- In the manuscript, the authors often refer to “production” workers. Are construction workers considered production workers? If they are, then maybe it is more relevant to use the work “construction” workers rather than “production” workers. In my mind, production workers are more aligned with manufacturing operations.

Reviewer 3 Report
the article approaches a rich and important subject, considering the target of study. The methodology is duly described, and the analysis tools used, despite being complex, demonstrate their usefulness for the study. As the only suggestion, attach a questionnaire to frame the questions selected to assess the dimensions under study.
Reviewer 4 Report
Dear Authors,
Thanks for giving me the opportunity to read this interesting paper. I have mainly one comment concerning the self-efficacy concept. It is mentioned that people with high safety self-efficacy level will have higher levels of confidence when faced risks. This is certainly well documented in the literature. However, from a practitioner point of view, too much confidence (too much self-efficacy) is also negative for safety. This has been largely shown through the cognitive bias literature. Some further explanations could be made.
Typos:
p.3 (no line number): Neal and Griffin[13] introuced
p.5 (no line number): leadership member exchang[
